# Treating Neural Image Compression via Modular Adversarial Optimization: From Global Distortion to Local Artifacts

## Abstract

The rapid progress in neural image compression (NIC) led to the deployment of advanced codecs, such as JPEG AI, which significantly outperform conventional approaches. However, despite extensive research on the adversarial robustness of neural networks in various computer vision tasks, the vulnerability of NIC models to adversarial attacks remains underexplored. Moreover, the existing adversarial attacks on NIC are ineffective against modern codecs. In this paper, we introduce a novel adversarial attack targeting NIC models. Our approach is built upon two core stages: (1) optimization of global-local distortions, and (2) a selective masking strategy that enhances attack stealthiness. Experimental evaluations demonstrate that the proposed method outperforms prior attacks on both JPEG AI and other NIC models, achieving greater distortion on decoded images and lower perceptibility of adversarial images. We also provide a theoretical analysis and discuss the underlying reasons for the effectiveness of our attack, offering new insights into the security and robustness of learned image compression.

## 1 Introduction

Recent advances in neural image compression (NIC) have led to state-of-the-art codecs, such as JPEG AI (Ascenso et al. (2023)), that significantly outperform traditional compression methods. However, the robustness of these learned compression systems has not received much attention. Previous studies have shown that small perturbations, which are imperceptible to humans, can significantly degrade NIC outputs. For example, adding a small amount of noise can cause "severe distortion" in the decoded image or substantially increase the compressed bit rate. In other words, neural image compression models are vulnerable to adversarial attacks. Most existing attacks target a single global metric, such as PSNR, and employ simple methods to identify adversarial perturbations, which have been developed for other types of attacks, rather than specifically for NIC. While these methods expose the vulnerability of NIC, they don't create visible artifacts often enough and remain relatively ineffective against modern codecs. They also highlight the gap: current adversarial methods for NIC do not fully exploit features of the image compression task and the opportunity to craft localized distortions. In this work, we take a comprehensive approach to adversarial attacks on neural compression. We introduce an attack that targets both global quality and local artifacts explicitly. Crucially, our attack operates as a modular adversarial optimization. At each iteration, it chooses among different objectives (global or local) and gradient updates strategies (signed or normalized), and then projects the perturbation back under the norm constraint. This modular design makes the optimization both flexible and effective. We further enhance stealth by applying a selective frequency-domain mask. The novelty in the design of the proposed approach enabled it to achieve a new state-of-the-art level of attack efficiency, including attacks on modern NIC methods.

## 2 Related Work

**Neural image compression.** Neural image compression has been developing rapidly in recent years. Ballé et al. (2016) published the foundation for many subsequent models. They proposed a generalized divisive normalization as a core nonlinear transform in NIC, followed by uniform scalar quantization. Agustsson et al. (2017) presented a soft-to-hard vector quantization approach

within a compressive autoencoder architecture, enhancing end-to-end learning by bridging the gap between differentiable and non-differentiable quantization. Later, Ballé et al. (2018) formulated image compression as a variational inference problem, introducing hyperpriors to model the entropy more effectively and improve compression rates. Several improvements were proposed for entropy modeling, including the extension of the hierarchical Gaussian scale mixture (GSM) model to a Gaussian mixture model (Minnen et al. (2018)) and the discretization of Gaussian mixture likelihoods combined with a simplified attention mechanism (Cheng et al. (2020)). For enhancing image reconstruction and fidelity,Mentzer et al. (2020) proposed a GAN-based compression method, and Yang & Mandt (2024) proposed a scheme that utilizes an encoder to map images onto a contextual latent variable, which is then fed into a diffusion model for reconstructing the source images. He et al. (2022) describes a new ELIC model that adopts stacked residual blocks as nonlinear transforms and uses the Space-Channel ConTeXt (SCCTX) model, which is a combination of the spatial context model and the channel conditional backward-adaptive entropy model. Zou et al. (2022) introduces a flexible window-based attention module to enhance image compression models and trains CNN and Transformer models that reach promising results. Liu et al. (2023a) proposes using parallel transformer-CNN mixture blocks to combine the advantages of both approaches, as well as a new entropy model that uses a SWIN-transformer-based attention module with channel squeezing. Duan et al. (2023) adopted a hierarchical VAE architecture, called ResNet VAE, for image compression, using a uniform posterior and a Gaussian convolved with a uniform prior. Wang et al. (2023) creates a real-time neural image compression model using residual blocks and depth-wise convolution blocks, and then uses mask decay and novel sparsity regularization loss to transfer knowledge to smaller models.

**Attacks on NICs.** Chen & Ma (2023) shows that NIC models are severely vulnerable to adversarial perturbations, which decrease the quality of the reconstructed image after compression. They adapted the I-FGSM (Kurakin et al. (2017)) attack to reduce the quality of the compressed image. They proposed the Fast Threshold-constrained Distortion Attack (FTDA), which is a more efficient attack on images after compression. Liu et al. (2023b) introduced an adversarial attack on NIC models based on I-FGSM, focusing on the bitrate aspect. They examined the effects of this attack on various codec architectures and found that their factorized attention model was the most resistant to the weight gain of the compressed representation. Wu (2024) propose a practical paradigm of Specific-ratio Rate-Distortion Attack (SRDA) (together with Agnostic-ratio RD-Attack (ARDA)) to assess the robustness of NIC methods at a target bitrate level. The authors also introduce two analytical tools — Entropy Causal Intervention and Layer-wise Distance Magnify Ratio — to localize vulnerable codec components. They demonstrate that hyperprior significantly increases bitrate under attack, and the IGDN block enhances input disturbances.

## 3 PROBLEM FORMULATION

An attack on a neural image codec that reduces decoded image quality can be formulated as a constrained adversarial optimization. Given an original image $I$, we seek a small perturbation $\delta$ (e.g., with $|\delta|_p \leq \varepsilon$) such that the difference between the adversarial image $I^* = I + \delta$ and its compressed-and-decompressed version is maximized. In practice, quality loss is measured by full-reference metrics like MSE, PSNR or SSIM. Thus we maximize a distortion function $d(I^*, \hat{I}^*)$ (e.g. MSE or $1 - \text{SSIM}(x, \hat{x})$, etc.) where $\hat{I}^*$ is the reconstructed image after compression of $I^*$. Formally, the attack can be written as:

$$\underset{I^*:|I^*-I|_p \leq \varepsilon}{\arg\max} \quad d\big(I^*, C(I^*)\big),$$

where $C(\cdot)$ denote the complete encoding-decoding process. In other words, we add an imperceptible targeted noise to $I$ so that the codec's output image has significantly lower quality.

## 4 PROPOSED METHOD

To search for adversarial perturbations, we propose an iterative gradient-based method under $l_\infty$ norm constraint. Since the distortion function $d(\cdot)$ may not always be differentiable or may be easily hacked during an attack, it may also be challenging to optimize. Additionally, we want the found perturbation to be effective for different distortion functions. Therefore, instead of using the

---

**Algorithm 1 NOpt** (Norm optimisation algorithm)

---

**Input**: image $I$, adversarial image $I^*$, parameters: $\varepsilon$, norm optimisation step $\beta$, maximum iterations for norm optimisation $M$
**Output**: bounded adversarial image $I_b^*$

1: Let $I_b^* = I^*$.
2: Let $err = L(I, I_b^*)$
3: Let $iteration = 0$
4: **while** $err > 0$ and $iteration < M$ **do**
5: $\quad I_b^* = I_b^* - \beta * sign(\nabla err)$
6: $\quad$ Let $err = L(I, I_b^*)$
7: $\quad iteration = iteration + 1$
8: **end while**
9: **return** $I_b^*$

---

distortion function during the attack, we suggest calculating and taking the gradient of a simpler auxiliary function $f(I, I^*, C(I), C(I^*))$, which is more responsive to image quality degradation and compression artifacts. This auxiliary function can be different for each iteration of the attack and is referred to as the attack objective. Thus, at each iteration, a step is taken to solve the following optimization problem:

$$\arg\max_{I^*:|I^*-I|_\infty \leq \varepsilon} f\big(I, I^*, C(I), C(I^*)\big)$$

Each iteration of our method follows a modular structure: first, we select the attack objective and calculate its gradient. Then, we take a gradient-based update in the chosen direction to maximize that objective. Finally, we project the result to ensure that the perturbation respects the constraint (norm optimization). This modular approach makes the attack both flexible and controllable.

### 4.1 $l_\infty$ NORM OPTIMISATION ALGORITHM

To maintain the constraint after the optimization step of the target attack function $f(\cdot)$, we propose an NOpt (Norm optimisation) algorithm (Algorithm 1) based on the I-FGSM method with the target function:

$$L(I, I^*) = \sum_{x \in I - I^*} \mathbb{I}(|x| > \varepsilon) * |x|,$$

where $x$ denote components of the vector $I - I^*$. This allows us to switch from optimizing the $l_\infty$ norm to the $l_1$ norm, which has a less sparse gradient and accelerates the convergence process. The algorithm iterates until either the norm limit is reached or the maximum number of iterations is exceeded.

### 4.2 FLEXIBLE OPTIMISATION DIRECTION

**Signed gradient direction.** The use of the sign of each component of the gradient (as in the I-FGSM) ensures an $l_\infty$-bounded step size. The update for each coordinate is exactly $\pm lr$ (where $\pm lr$ denotes a learning rate), thus keeping the perturbation within the specified bounds with low cost. The downside of using sign-based updates is that they lose the relative magnitude information contained in the gradient. This means that all input dimensions are treated equally, ignoring any directional nuances in the loss landscape. As a result, the direction of the update becomes aligned with the axes and discontinuous, resulting in coarse approximations of the true gradient path. This can lead to suboptimal convergence, especially when the loss function has an anisotropic shape. In such cases, sign-based methods may take inefficient paths, oscillate, or miss opportunities to exploit informative gradient directions that are shallow but important. In practice, this limitation can lead to:

*Slower convergence*: The attack may require more iterations to reach the desired level of distortion, as it does not follow the optimal path.

*Wasted perturbation budget*: Pixels may be unnecessarily perturbed to the maximum allowed extent ($\varepsilon$), even if they do not contribute significantly to the objective.

*Blind spots in optimization*: Fine-grained local structures in the loss surface (e.g., near-flat valleys) may be missed, as the step size is effectively binary per pixel and cannot capture complex variations in the loss landscape.

**Normalized gradient direction.** To address these issues, we suggest computing a vector with a standard distribution based on the gradient direction by subtracting the average of its components and scaling it by the standard deviation.

$$direction = normalize(\nabla f) = \frac{\nabla f - mean(\nabla f)}{std(\nabla f)}.$$

This process produces an update direction that is smooth, continuous, and well-suited to the landscape of the loss function. Unlike discrete $\pm 1$ directions used in sign methods, normalized directions maintain pixel-wise proportionality, allowing more important pixels (in terms of sensitivity to loss) to receive larger proportional updates. This improves convergence and enables perturbation targeting regions that are sensitive to compression more effectively. In short, subtracting the mean removes global drift and dividing by standard deviation normalizes local contrast, producing a balanced and effective update — one that remains true to underlying gradients without being too aggressive or constrained by axis. The normalized step is particularly valuable for fine tuning or correcting rough steps taken by sign gradient.

At the same time, this transformation allows us to limit the absolute pixel change per step with some probability, thanks to the Chebyshev inequality. This gives us a probabilistic estimate of the number of iterations required for Algorithm 1 to meet the norm constraints after an attack step.

**Proposition.** Let's assume that, in iteration $t$, a normalized gradient direction $\overrightarrow{r}$ was used with learning rate $lr$ to optimize the target function of the attack. Then, when we use NOpt (Algorithm 1) with step size $\beta$ to limit perturbation, the number of NOpt's iterations $T$ has a following probabilistic estimation:

$$\mathbb{P}\big(T \geq \frac{lr * k}{\beta}\big) \leq 1 - (1 - \frac{1}{k^2})^{H*W},$$

where $H, W$ denote dimensions of the image $I$. The proof of the proposition can be viewed in the appendix.

This statement allows us to theoretically demonstrate the possibility of limiting the number of iterations without sacrificing the quality of NOpt algorithm in the majority of cases. In practice, the actual number of necessary iterations is often significantly lower, as the statement assumes the worst-case scenario where the largest value of step must be completely removed. However, this method of determining the direction of the next step still requires a significantly larger number of steps of the NOopt algorithm compared to the signed approach. This can lead to a significant blurring of the original direction, or, if the choice of M is sufficiently rigid, to the accumulation of errors. To solve this problem, we alternate between sign-based and normalized updates in order to get the best results from both approaches. A sign-gradient step immediately ensures the $l_\infty$-norm limit is enforced and keeps the perturbation stable, while a normalized-gradient step takes advantage of richer gradient information to accelerate progress. By alternating between these two update forms, we can combine their complementary strengths and mitigate their individual limitations. The sign step stabilizes the optimization, ensures norm constraints, and guarantees global coverage, while the normalized step provides fine-grained corrections, leveraging gradient curvature and structure. This interplay smooths out the individual weaknesses of both approaches. Where the sign direction lacks precision, the normalized step refines it. And where the normalized step is prone to instability, the sign step reinforces it. In essence, the alternation allowes the optimizer to explore and converge, while avoiding the characteristic errors of either method, leading to a more reliable and efficient optimization path, compared to either approach alone.

### 4.3 GLOBAL AND LOCAL OPTIMISATION MODULE

Our attack proceeds in two sequential module stages, targeting different distortion measures.

**Global quality degradation.** We first maximize the overall difference between the original and compressed images. In practice, we use the MSE over the entire image as our loss function.

$$f_{global}(I^*, C(I^*)) = MSE(I^*, C(I^*))$$

This step reduces compression quality uniformly by increasing the overall pixel-by-pixel error. Intuitively, maximizing the global MSE results in a significant decrease in image fidelity, establishing a strong baseline for distortion. It was at this stage of the global decline in image quality that all previously proposed attack methods ceased to be effective. However, as can be seen from examples, experiments, and other studies, the decrease in quality in practice is not uniform: local areas with severe distortions (artifacts) form. At the same time, optimization continues to be carried out using global quality metrics, which may not accurately reflect reality and may not lead to maximum degradation, as they focus on areas and pixels that are less susceptible to artifacts.

**Local quality drawdown.** Therefore, we aim to mitigate the local perceptual quality drop that occurs after the first stage. As shown Tsereh et al. (2024), the Structural Similarity Index Metric (SSIM) for the Y component in the YUV color space is more suitable for finding local distortions in neural compression. Therefore, we compute two local SSIM similarity maps: one between the original image and the compressed image $\texttt{SSIM}(I, C(I))$, and another between the adversarial image and the compressed adversarial image $\texttt{SSIM}(I^*, C(I^*))$. By calculating the difference $\Delta(I, I^*, C(I), C(I^*)) \in \mathbb{R}^{H \times W}$ between these two maps, we can identify areas with the maximum quality drop caused by the attack. To smooth outliers, we also use average pooling with a kernel size of 32.

$$\Delta = AvgPool_{32}\big(\texttt{SSIM}(I, C(I)) - \texttt{SSIM}(I^*, C(I^*))\big)$$

Let's examine the single map between the adversarial images. We may mistakenly identify areas that have decreased in quality due to features in the original image rather than the attack. To detect distortions in various parts of an image, $\Delta$ is divided into non-overlapping blocks of a specific size $P \times P$. Each block has a maximum value, which, with proper selection of $P$, allows us to locate and enhance many artifacts in the image while ignoring relatively stable areas. The sum of these maximum intensities is used as an objective function:

$$f_{local}(I, I^*, C(I), C(I^*)) = \sum_{i=1}^{\left\lceil \frac{H}{P} \right\rceil} \sum_{j=1}^{\left\lceil \frac{W}{P} \right\rceil} \max_{(u,v) \in \mathcal{B}_{ij}} \Delta(u, v), \text{where}$$

$$\mathcal{B}_{ij} = \{(u, v) \mid u \in [P(i-1), Pi), \ v \in [P(j-1), Pj)\}.$$

The sequential order of the two stages is essential. By first reducing the global quality, we lower the overall image fidelity, making it easier for the second stage to focus on specific artifacts. The global stage ensures a significant drop in PSNR (high MSE), while the local stage sharpens and localizes distortions (low SSIM). This two-stage approach is more effective than a single combined objective, as it allows for dedicated optimization of each stage's goals. Therefore, the final attack is the sequential application of the Algorithm 2, first with the objective function $f_{global}$ and then with function $f_{local}$.

## 4.4 Fidelity improvement by frequency module

To make the attack less noticeable, we remove low-frequency components from the adversarial perturbation $\delta = I^* - I$ after applying the attack algorithm. Specifically, we transform the noise into the frequency domain (for example, using the DCT or FFT) and set all components below a certain threshold $R$ to zero.

$$\delta_{\text{high}} = cut(\delta, R) = \mathcal{F}^{-1}\big(Mask(R) \cdot \mathcal{F}(\delta)\big)$$

$$Mask(R)(u, v) = \begin{cases} 0 & \text{if } \sqrt{(u - H/2)^2 + (v - W/2)^2} < R \\ 1 & \text{otherwise} \end{cases}$$

where $\mathcal{F}(\cdot)$ and $\mathcal{F}^{-1}(\cdot)$ denote the forward and inverse 2D Fourier (or DCT) transform applied per channel High-frequency noise is usually harder for the human eye to detect, so by limiting the perturbation to high frequencies (i.e., applying a low-frequency constraint), we preserve the perceptual similarity with the original image. This approach avoids creating large artifacts or obvious color gradients, producing instead subtle high-frequency speckles that make the adversarial image look more like the original one. The threshold $R$ is chosen individually for each image in a discrete set $\{0, 10, 20, ..., 100\}$, such that the relative reduction in the effectiveness of the attack (measured by

---

**Algorithm 2** Attack algorithm

---

**Input**: codec $C$, image $I$, attack objective function $f$, parameters: $\varepsilon$, $lr$, $N$
**Output**: adversarial image $I^*$

1: Let $I^* = I$, $iteration = 0$.
2: // Attacking the objective function
3: **while** $iteration < N$ **do**
4:     $G = \nabla f(I, I^*, C(I), C(I^*))$
5:     // Alternate the method of selecting the direction of optimization step
6:     **if** $iteration \equiv 1 \mod 2$ **then**
7:         $direction = sign(G)$
8:     **else**
9:         $direction = normalize(G)$
10:     **end if**
11:     // Optimization step
12:     $I^* = I^* + lr * direction$
13:     // Use Algorithm 1 to limit the norm of adversarial perturbation
14:     $I^* = NOpt(I, I^*, \varepsilon)$
15:     $iteration = iteration + 1$
16: **end while**
17: **return** $I^*$

---

the value of the objective function of the second stage $f_{local}$) is less than the common threshold value $Trh$:

$$R = \max \left\{ R' \in \{0, 10, 20, ..., 100\} : Per(R') \le Trh \right\},$$

$$Per(R') = \frac{f_{local}(I^*) - f_{local}(I + cut(\delta, R'))}{f_{local}(I^*)}.$$

## 5 EXPERIMENTS

To thoroughly assess the proposed attack method, we compare it to previous methods by attacking a variety of different NICs. We employ several approaches to evaluate the performance of the attacks and measure their invisibility.

### 5.1 NIC MODELS

We evaluate a wide range of neural image compression (NIC) models, spanning from classic hyperprior architectures to more recent transformer-based codecs. These include Ballé et al. (2018) factorized and hyperprior models, Minnen et al. (2018) hierarchical model. We also test the Cheng et al. 2020 variants ("anchor" and "self-attention"), the ELIC model of He et al. (2022), the HiFiC GAN-based model of Mentzer et al. (2020), the QRes-VAE of Zou et al. (2022), and the mixed Transformer-CNN LIC-TCM model of Liu et al. (2023a). Additionally, we include the new JPEG-AI standard (Ascenso et al. (2023)) (version 7.1), both in its Base Operation Point (BOP) and High Operation Point (HOP) modes. This standard also includes additional tools that can help improve the efficiency and adaptability of compression. For example, these include Residual and Variation Scale (RVS) or filters such as Cross-Color Filter (ICCI) or Luma Edge Filter (LEF). The specific set of tools and their settings may vary depending on the codec configuration. That's why we train attacked images for JPEG AI without specific tools (tools off), and evaluation will be based on default settings with and without these tools (tools on/tools off) for each operation point of the codec. For each codec, we run experiments at four different compression rates (or quality levels), covering both low and high bitrate scenarios.

### 5.2 ATTACKS

We compare our method with FTDA, I-FGSM, and SRDA attacks. Chen & Ma (2023) use the square error between the compressed images before and after the attack as the objective function

for their attack. For a more comprehensive comparison, we also evaluate their attack variants using $f_{global}$ as the objective function. The SRDA attack involves the possibility of restrictions based on different norms ($p = \infty$, $p = 2$). Since the FTDA limits are based on $p = 2$ and our method is based on $p = \infty$, both restriction variants are considered for SRDA. We also evaluate the effectiveness of our method without some modules: without a normalized direction and without using $f_{local}$. After testing the attacks, we found that for all attacks except for FTDA, 30 iterations were sufficient to converge. However, FTDA required 100 iterations in order to work correctly. In our method, the number of iterations was equally divided between the objective functions (the first 15 iterations with $f_{global}$ and the next 15 with $f_{local}$). Our work used the $\varepsilon = 7/255$ (and $\varepsilon = 0.0006$ for FTDA and SRDA ($l_2$)) perturbation budget and block $\mathcal{B}_{ij}$ with size $P = 75$ (experiments with an alternative perturbation budget and different variations of the $P$ parameter can be viewed in the appendix).

## 5.3 DATASETS

We test attacks performance and the robustness of models on several standard image datasets. The KODAK PhotoCD dataset Kodak (1991) (24 high-quality uncompressed images) is used as a classic evaluation suite. In addition, we use 100 images from the Cityscapes Cordts et al. (2016) urban street scene dataset, as well as a random subset of 50 images from the NIPS 2017: Adversarial Learning Development Set Competition Page. These datasets cover both general photographic content (KO-DAK and NIPS) and domain-specific scenes (Cityscapes), and are commonly used for compression evaluations.

## 5.4 METHODS FOR EVALUATING THE PERFORMANCE OF ATTACKS

We employ two complementary evaluation methods.

*Quality drop metric.* For each codec and attack, we measure the degradation in reconstruction quality caused by the attack. We use a selected quality metric $Q$, such as PSNR, MS-SSIM Wang et al. (2003) or VMAF Li et al. (2018), to calculate the difference between the quality after compression of clear and attacked images. This difference is then divided by the quality of the clear image after compression to obtain a relative drop.

$$QDM = \frac{Q(I, C(I)) - Q(I^*, C(I^*))}{Q(I, C(I))}$$

We average the $QDM$ over all test images and all four bitrates for each codec, to get an average relative drop in quality ($\widetilde{QDM}$).

*Compression-efficiency drop metric.* To quantify how attacks affect compression efficiency, we compared each NIC to a JPEG2000 baseline using the BSQ-rate measure (Zvezdakova et al. (2020)). Specifically, we first compute a "bit savings" vs quality (BSQ-rate) curve for the NIC and JPEG2000 on a clean dataset. We repeat the comparison on attacked images and report the percentage increase in the BSQ-rate metric caused by the attack (an increase indicates worse efficiency). This captures the relative drop in compression performance under attack, including changes in bit-rate. This approach requires careful filtering of examples and is not meaningful for codecs that completely fail under attack, but it provides a more direct measure of compression degradation across rates.

$$CEDM = \frac{\text{BSQ-rate}(I^*) - \text{BSQ-rate}(I)}{\text{BSQ-rate}(I)}.$$

This metric is calculated only for those images whose attacked versions have an overlap of at least 25% between the ranges of the RD-curves for the neural network and traditional codecs. We average the $CEDM$ over all test images for each codec, to get an average compression-efficiency drop ($\widetilde{CEDM}$).

*Attack visibility.* We also evaluate the visibility of the adversarial perturbation by $Q(I, I^*)$. We average the $Q(I, I^*)$ over all test images and all four bitrates for each codec, to get an average attack visibility ($\widetilde{Q}$). To make the visibility metric relative and independent of the range of quality metric $Q(\cdot)$, the values of the visibility metric are normalized for each codec and increased by 2 to

eliminate negative values:

$$VM = \frac{\widetilde{Q} - \text{mean } \widetilde{Q} \text{ by attacks}}{\text{std } \widetilde{Q} \text{ by attacks}} + 2.$$

*Generalized assessment of attack performance.* Since a large number of codecs are involved in the comparison, for the final assessment of the effectiveness of attacks, performance metrics (CEDM, QDM) and visibility metrics are averaged over all codecs, and the final estimates are determined as the square root of the product of the average performance metric and visibility metric:

$$QD\text{-}SCORE = \sqrt{VM \cdot \widetilde{QDM}} \quad \text{and} \quad CED\text{-}SCORE = \sqrt{VM \cdot \widetilde{CEDM}}.$$

We also measure metrics $QD\text{-}SCORE$ and $CED\text{-}SCORE$ separately for the JPEG AI codec family, as well as for each codec (results for each codec can be viewed in the appendix).

## 6 RESULTS AND DISCUSSION

**Overall Attack Effectiveness.** Table 1 demonstrates that the proposed modular adversarial attack outperforms baseline methods (I-FGSM, FTDA, and SRDA) across a diverse set of NIC architectures. In terms of quality degradation (QD-SCORE), our method achieves the most significant relative drop in PSNR, VMAF, and MS-SSIM, while maintaining low perceptibility of perturbations. Similarly, in compression efficiency degradation (CED-SCORE), our approach leads to substantially higher bit-rate inefficiency compared to competing attacks, confirming that it can destabilize both reconstruction quality and rate–distortion behavior.

| Attack | $QD\text{-}SCORE$ | | | $CED\text{-}SCORE$ | | |
|---|---|---|---|---|---|---|
| | PSNR | VMAF | MS-SSIM | PSNR | VMAF | MS-SSIM |
| OUR + freq. module | **0.59** | 0.45 | **0.42** | 0.81 | 0.45 | **0.79** |
| OUR | 0.57 | **0.52** | 0.40 | 0.82 | **0.59** | 0.65 |
| OUR w/o $f_{local}$ | 0.54 | 0.46 | 0.32 | 0.75 | 0.52 | 0.70 |
| OUR w/o normalized direction | 0.54 | 0.45 | 0.31 | 0.78 | 0.50 | 0.63 |
| I-FGSM with $f_{global}$ | 0.53 | 0.48 | 0.27 | 0.57 | 0.54 | 0.53 |
| FTDA with $f_{global}$ | 0.56 | 0.46 | 0.37 | **0.85** | 0.57 | 0.77 |
| I-FGSM | 0.38 | 0.13 | 0.26 | 0.36 | 0.19 | 0.42 |
| FTDA | 0.43 | 0.17 | 0.31 | 0.66 | 0.22 | 0.71 |
| SRDA ($l_2$) | 0.41 | 0.20 | 0.18 | 0.65 | 0.20 | 0.51 |
| SRDA ($l_\infty$) | 0.37 | 0.10 | 0.24 | 0.29 | 0.29 | 0.49 |

Table 1: Comparison of attack performance across all tested NIC codecs. QD-SCORE (quality degradation) and CED-SCORE (compression efficiency degradation) are reported for PSNR, VMAF, and MS-SSIM. The proposed method consistently achieves higher degradation with lower visibility than baseline attacks.

**Robustness Across Codecs.** Our attack generalizes well across codecs, from classical hyperprior and hierarchical models to modern transformer-based codecs and GAN-based HiFiC. As shown in Table 2, its performance on JPEG AI is particularly noteworthy, as prior methods were comparatively ineffective. Both Base Operation Point (BOP) and High Operation Point (HOP) configurations were significantly degraded, indicating that the proposed optimization strategy can penetrate even the most advanced codec designs. Fig. 2 and Fig. 3 in the Appendix contain more information.

**Contribution of Individual Modules.** The ablations in Tables 1 and 2 highlight the importance of each module. Removing the local distortion stage (w/o $f_{local}$) reduces both QD-SCORE and CED-SCORE, underscoring that local artifact amplification complements global degradation. Likewise, excluding the normalized direction update diminishes attack strength, showing that alternation between signed and normalized gradients provides a more balanced and efficient optimization path. Incorporating the frequency-domain masking (freq. module) improves imperceptibility while preserving attack effectiveness, yielding the best trade-off between strength and stealth.

| Attack | QD-SCORE | | | CED-SCORE | | |
|---|---|---|---|---|---|---|
| | PSNR | VMAF | MS-SSIM | PSNR | VMAF | MS-SSIM |
| OUR + freq. module | **0.218** | 0.078 | 0.081 | **0.517** | 0.202 | **0.514** |
| OUR | 0.192 | **0.092** | **0.087** | 0.422 | 0.209 | 0.376 |
| OUR w/o $f_{local}$ | 0.153 | 0.091 | 0.085 | 0.355 | **0.232** | 0.337 |
| OUR w/o normalized direction | 0.152 | 0.087 | 0.084 | 0.348 | 0.164 | 0.318 |
| I-FGSM with $f_{global}$ | 0.138 | 0.079 | 0.066 | 0.218 | 0.052 | 0.228 |
| FTDA with $f_{global}$ | 0.165 | 0.079 | 0.078 | 0.325 | 0.028 | 0.280 |
| I-FGSM | 0.087 | NaN | 0.046 | 0.109 | 0.045 | 0.293 |
| FTDA | 0.071 | NaN | 0.045 | 0.132 | 0.054 | 0.372 |
| SRDA ($l_2$) | 0.123 | 0.062 | 0.070 | 0.232 | 0.081 | 0.286 |
| SRDA ($l_\infty$) | 0.113 | 0.013 | 0.078 | 0.104 | 0.061 | 0.218 |

Table 2: Attack performance on the JPEG AI codec family. Despite its advanced design, JPEG AI is highly vulnerable to the proposed attack, with significant drops in both reconstruction quality (QD-SCORE) and compression efficiency (CED-SCORE). The frequency-domain masking variant provides the best trade-off between stealth and strength.

**Visual Assessment.** Fig. 1 shows decompressed images after different attacks. Competing methods (I-FGSM, FTDA) introduce visible distortions; however, the distortions that occur after our attack are more pronounced. Our method provides effective perturbations that yield pronounced artifacts post-compression while leaving adversarial inputs nearly indistinguishable from the originals. For JPEG AI, the compared attacks are ineffective, as they do not produce visible artifacts.

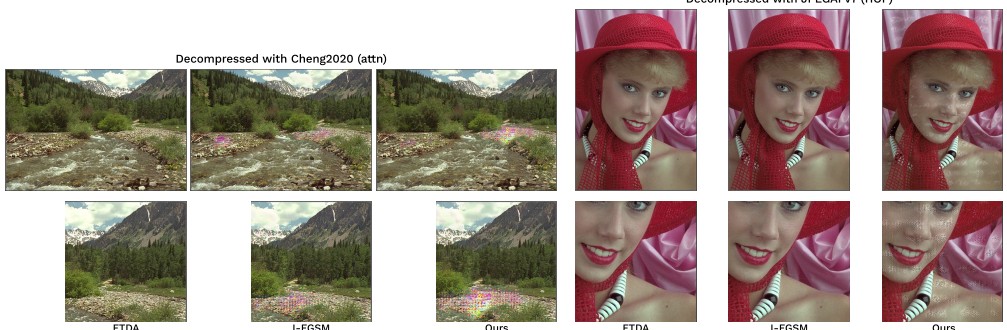

Figure 1: Adversarial images generated by our proposed attack and compared to baseline methods, compressed and decompressed using two different codecs. Our method produces more pronounced artifacts post-compression while maintaining adversarial inputs that remain visually similar to the original images, unlike competing attacks, which often fail to create significant artifacts.

## 7 CONCLUSION

In this work, we introduced a modular adversarial attack framework for neural image compression, combining global distortion maximization, local artifact amplification, and frequency-domain masking. Our results across a wide range of codecs, including JPEG AI, demonstrate that this approach achieves state-of-the-art quality and efficiency degradation compared to existing attacks, while maintaining imperceptible perturbations. The global and local strategies are critical, as global degradation reduces the baseline quality while local artifact optimization enhances the perceptual and structural impact of the distortions. The frequency masking module improves stealth by reducing low-frequency artifacts, which would otherwise be more noticeable, without sacrificing attack strength. These findings reveal a critical vulnerability in modern NIC systems, indicating that compression methods remain highly susceptible to perturbations. Our analysis also provides insights that may guide the design of future resilient codecs and motivate the development of robust training or defense mechanisms against adversarial manipulations.

## 8 REPRODUCIBILITY STATEMENT

We attach the full code for the experiments, and a Docker image in the supplementary and on a public repository upon acceptance.

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

# A  APPENDIX

## A.1  PROOF OF THE PROPOSITION

Let $l_\infty^t = \|I_t^* - I\|$ (norm of perturbation after $t$ iteration), where $I_t^*$ denote adversarial image after $t$ iteration. Then, since NOpt uses a sign-direction approach, for sufficiently small $\beta$, the number of iterations can be easily approximately calculated as:

$$T = \begin{cases} \frac{l_\infty^t - \varepsilon}{\beta}, \text{if } l_\infty^t > \varepsilon \\ 0, \text{else} \end{cases} \quad .$$

Let $x$ denote the element of the vector $r$ with the maximum absolute value. Then, the evaluation $l_\infty^t \leq \varepsilon + |x| * lr$ is valid. Therefore, inequality $l_\infty^t \geq \varepsilon + k * lr$ can only occur if inequality $|x| \geq k$ holds. Let's write Chebyshev's Inequality for the random variable $y$ (random element of $r$), taking into account that its expected value is 0 and its standard deviation is 1:

$$\mathbb{P}\big(|y| \geq k\big) \leq \frac{1}{k^2}$$

Then it is true for $x$: $\mathbb{P}\big(|x| \geq k\big) \leq 1 - (1 - \frac{1}{k^2})^{H*W}$ From which we get

$$\mathbb{P}\big(l_\infty^t \geq \varepsilon + k * lr\big) \leq 1 - (1 - \frac{1}{k^2})^{H*W}. \ \square$$

## A.2  EFFECTIVENESS OF ATTACKS ON DIFFERENT CODECS.

In Fig. 2 and 3 are the results across different NIC architectures. They confirm that our attack is broadly effective and generalizes well beyond a single codec family. While traditional methods such as I-FGSM and FTDA show limited degradation—especially against stronger codecs like JPEG AI—our modular approach consistently produces higher QD-SCORE values. The inclusion of local artifact amplification and normalized gradient updates proves particularly important for more advanced models, enabling substantial quality degradation even where prior attacks fail. These findings highlight the robustness and versatility of the proposed method across both classic hyperprior models and modern transformer- or GAN-based codecs.

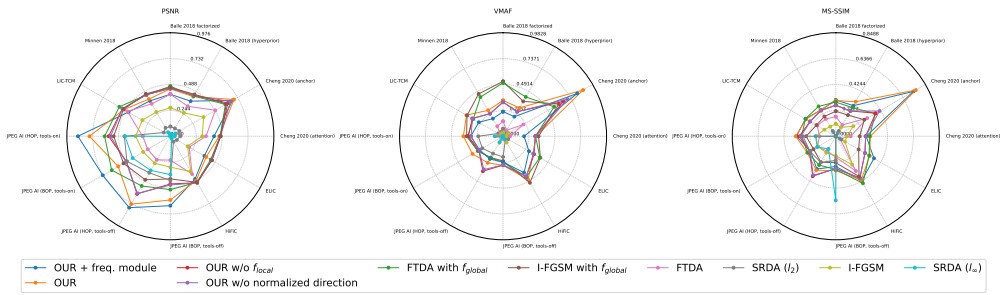

Figure 2: Comparison of QD-SCORE on different NIC models. Lines represent attack efficiency for the proposed method and other methods.

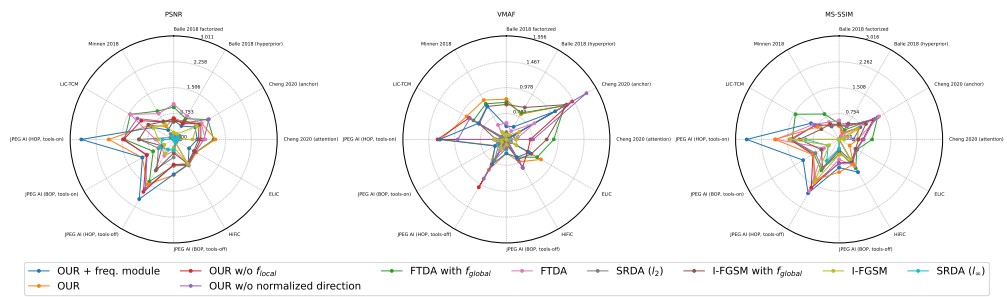

Figure 3: Comparison of CED-SCORE on different NIC models. Lines represent attack efficiency for the proposed method and other methods.

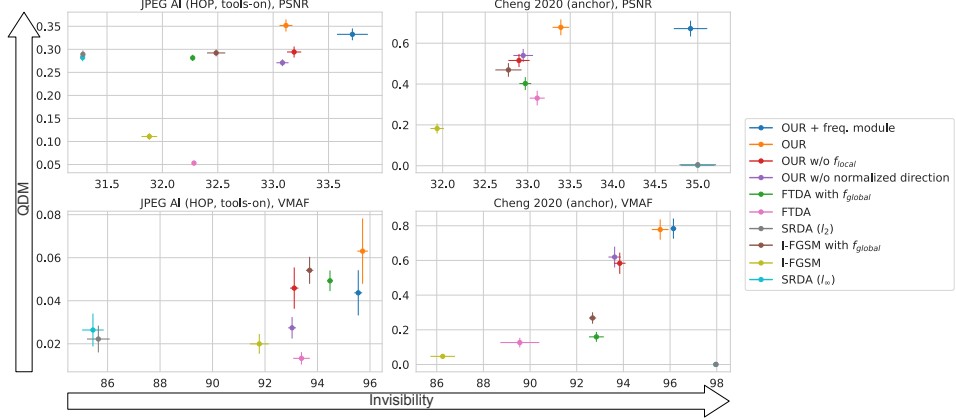

Figure 4: Comparison of QDM across different NIC models (with the lowest compression ratio in the their families). Our proposed method, particularly with the frequency-domain masking module, consistently achieves higher QDMs while maintaining stealth, outperforming baseline attacks across both classical and modern codecs.

## A.3 ATTACK INVISIBILITY-EFFICIENCY TRADE-OFF.

Fig. 4 illustrates the trade-off between invisibility and compression efficiency degradation (CED-SCORE). The results show that our proposed method consistently achieves higher CED-SCORE values at comparable levels of invisibility than baseline attacks, indicating a more effective balance between stealth and destructive impact. In particular, the frequency-domain masking variant attains the best compromise: it maintains imperceptible perturbations while still inducing significant efficiency losses in the codecs. This demonstrates that the modular design not only enhances attack strength but also improves its practicality by minimizing visual detectability.

## A.4 ALTERNATIVE PERTURBATION BUDGET.

In this section, we show the results of an experiment on KODAK PhotoCD dataset with a smaller perturbation budget $\varepsilon = 5/255$ (and $\varepsilon = 0.0003$ for FTDA and SRDA ($l_2$)). The results in the Tables 5 show that with a lower $\varepsilon$ attacks become much less effective, but our methods still show the best results.

| Attack | QD-SCORE | | CED-SCORE | |
|---|---|---|---|---|
| | PSNR | VMAF | PSNR | VMAF |
| OUR + freq. module | **0.17** | 0.1 | **0.23** | 0.11 |
| OUR | 0.13 | **0.12** | 0.17 | **0.14** |
| I-FGSM with $f_{global}$ | 0.10 | 0.1 | 0.11 | 0.12 |
| FTDA with $f_{global}$ | 0.12 | 0.9 | 0.18 | 0.11 |
| I-FGSM | 0.06 | 0.02 | 0.07 | 0.03 |
| FTDA | 0.08 | 0.05 | 0.1 | 0.06 |

Table 3: Comparison of attack performance across all tested NIC codecs on KODAK PhotoCD dataset with perturbation budget $\varepsilon = 5/255$. QD-SCORE (quality degradation) and CED-SCORE (compression efficiency degradation) are reported for PSNR, VMAF. The proposed method consistently achieves higher degradation with lower visibility than baseline attacks.

## A.5 ADVERSARIAL IMAGES EXAMPLE.

In this section, we show adversarial images before compression that correspond to the examples shown in Fig. 1, so that we can assess the visibility of attacks.

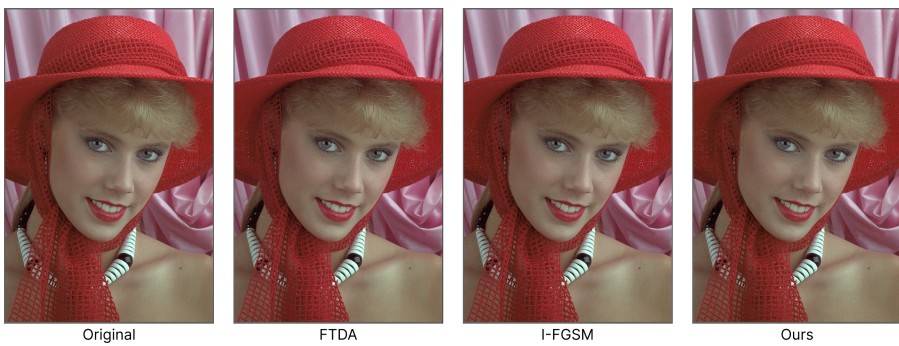

Figure 5: Adversarial images for JPEG AI v7 (HOP) generated by our proposed attack and compared to baseline methods before compression.

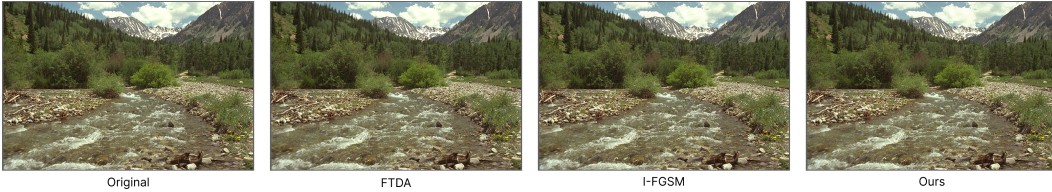

Figure 6: Adversarial images for Cheng2020 (attn) generated by our proposed attack and compared to baseline methods before compression.

## A.6 VARIATIONS OF THE $P$ PARAMETER.

The parameter P controls the trade-off between artifact localization granularity and computational efficiency. Empirical findings during development showed that $P \in [50, 100]$ approximately the

same results are obtained. It makes no sense to increase the parameter, since the current size is sufficient for local artifacts to occur, and increasing it will only lead to the loss of some of them. We selected $P = 75$ as the midpoint providing optimal balance.

| $P$ | QD-SCORE | |
|---|---|---|
| | PSNR | VMAF |
| 25 | 0.157 | 0.087 |
| 50 | 0.189 | 0.090 |
| 75 | 0.188 | 0.090 |
| 100 | 0.188 | 0.091 |
| 125 | 0.181 | 0.088 |
| 150 | 0.177 | 0.089 |

Table 4: The effectiveness of our attack at different P values on the JPEG AI codec family on the KODAK dataset.

## A.7 AN ATTACKS FOR DIFFERENT BITRATES.

| Attack | Quality = 0 | | | Quality = 1 | | |
|---|---|---|---|---|---|---|
| | VMAF | MS-SSIM | PSNR | VMAF | MS-SSIM | PSNR |
| OUR + freq. module | 0.23 | 0.35 | **0.45** | 0.15 | 0.26 | **0.66** |
| OUR | **0.38** | **0.36** | 0.37 | **0.31** | **0.33** | 0.48 |
| OUR w/o $f_{local}$ | 0.28 | 0.30 | 0.35 | 0.17 | 0.26 | 0.49 |
| OUR w/o normalized direction | 0.37 | 0.35 | 0.37 | 0.30 | 0.29 | 0.48 |
| I-FGSM with $f_{global}$ | **0.38** | 0.24 | 0.37 | **0.31** | 0.23 | 0.46 |
| FTDA with $f_{global}$ | **0.38** | 0.32 | 0.44 | **0.31** | 0.28 | 0.59 |
| I-FGSM | 0.00 | 0.17 | 0.18 | 0.03 | 0.18 | 0.29 |
| FTDA | 0.00 | 0.15 | 0.14 | 0.03 | 0.16 | 0.24 |
| SRDA ($l_2$) | 0.31 | 0.36 | 0.33 | 0.27 | 0.30 | 0.44 |
| SRDA ($l_\infty$) | 0.00 | 0.18 | 0.22 | 0.07 | 0.14 | 0.43 |
| Attack | Quality = 2 | | | Quality = 3 | | |
| | VMAF | MS-SSIM | PSNR | VMAF | MS-SSIM | PSNR |
| OUR + freq. module | 0.37 | **0.33** | **0.97** | 0.36 | **0.32** | **1.10** |
| OUR | **0.40** | 0.29 | 0.82 | **0.44** | 0.31 | 1.00 |
| OUR w/o $f_{local}$ | NaN | NaN | NaN | 0.32 | 0.28 | 0.93 |
| OUR w/o normalized direction | NaN | NaN | NaN | 0.25 | 0.25 | 0.87 |
| I-FGSM with $f_{global}$ | 0.34 | 0.21 | 0.55 | 0.36 | 0.26 | 0.77 |
| FTDA with $f_{global}$ | 0.30 | 0.24 | 0.68 | 0.36 | 0.25 | 0.70 |
| I-FGSM | 0.16 | 0.16 | 0.40 | 0.19 | 0.11 | 0.37 |
| FTDA | 0.12 | 0.12 | 0.30 | 0.18 | 0.06 | 0.31 |
| SRDA ($l_2$) | 0.23 | 0.23 | 0.53 | 0.05 | 0.14 | 0.38 |
| SRDA ($l_\infty$) | 0.21 | 0.22 | 0.54 | 0.04 | 0.13 | 0.37 |

Table 5: Comparison of attack performance (QD-SCORE) across the JPEG AI v7 (HOP) with different bitrates.

