# OpenReview forum: "Treating Neural Image Compression via Modular Adversarial Optimization: From Global Distortion to Local Artifacts"
_ICLR.cc/2026/Conference — Submitted to ICLR 2026_

### Official Review · Reviewer_vRSh · 2025-10-30

**Soundness:** 2
**Presentation:** 2
**Contribution:** 2
**Rating:** 4
**Confidence:** 3

**Summary:**

This paper introduces a novel adversarial attack against neural image compression (NIC) models. Compared to existing attacks, the proposed attack optimizes both global and local distortions and leverages a selective frequency-domain mask to enhance attack stealthiness.

**Strengths:**

- The authors provided a thorough review of the literature on NIC and its attack.
- The proposed attack is effective against modern codecs such as JPEG AI.

**Weaknesses:**

- The paper's introduction fails to clearly motivate the significance of its local optimization module, which is a key claimed contribution. The authors are suggested to move the justification from Section 4.3 to the introduction to immediately clarify why this local attack is necessary and effective.

- The proof of the proposition implicitly assumes independence across all pixels in the image, which is likely to be violated for natural images where pixels are spatially correlated. Additionally, the implication of this result on practical attacks is unclear, as efficiency is not typically a primary bottleneck for adversarial attacks.

- The general idea of using the frequency domain to improve an attack's stealthiness is not novel. It's a well-established concept in adversarial attack research (e.g., [1,2]).

- The l_\infty distortion bound used in experiments is not specified. Furthermore, it lacks an ablation study on this budget, making it hard to assess how the attack's performance scales at different perturbation levels.

[1] Luo, Cheng, et al. "Frequency-driven imperceptible adversarial attack on semantic similarity." Proceedings of the IEEE/CVF conference on computer vision and pattern recognition. 2022.
[2] Qin, Yao, et al. "Imperceptible, robust, and targeted adversarial examples for automatic speech recognition." International conference on machine learning. PMLR, 2019.

**Questions:**

- The performance gap between optimizing with and without normalized direction seems to be small. Would the signed gradient method achieve similar performance if optimized for more iterations?

- Does the generated adversarial perturbation transfer across different codecs?

- What is the perurabtion budget used in the experiments?

---

> ### Author Response · Authors · 2025-11-29
>
> Thank you for your detailed review of our work. We have carefully considered all your comments and questions, and we would like to provide you with detailed answers. In this response, we will address each of your points, explaining the contribution positioning, the theoretical justification, the frequency modulus, and the experimental details.
>
>
> $\textbf{W1:}$
>
> We agree this is an important contribution that deserves prominent positioning. In the camera-ready version, we will add a detailed description of our contributions to the introduction. We would also like to emphasize that this is not the only significant contribution of the work. Equally significant is the new approach to selecting the direction of optimization in Section 4.2.
>
> $\textbf{W2:}$
>
> We appreciate this technical critique. Let us clarify the Proposition’s role and address each concern:
>
> 1. On Natural Image Correlation:
> The reviewer is correct in pointing out that natural images do not follow the assumption of pixel independence. However, our method does not require independence between pixels, but rather between all the elements of the gradient vector. This is a more realistic assumption, as we are working with large and complex models for low-level tasks that do not require identifying global semantic information in images. And therefore the gradient often consists of high-frequency noise with little correlation between its elements. Additionally, this proposal explicitly uses worst-case reasoning and provides a probabilistic upper bound, rather than making assumptions about the actual behavior of the image. It should be noted that the lack of any correlations is simply part of this case. Therefore, our assessment remains valid and is merely approximate, which is sufficient for our purposes.
>
> 2. Why the Proposition Matters:
> The reviewer correctly points out that efficiency is not the main concern for adversarial attacks in general. However, the obtained proposition is essential for the theoretical soundness of the final design of the proposed attack algorithm. It simply ensures that all the necessary conditions for the attack are met. In practice, our method performs much better
>
>
> $\textbf{W3:}$
>
>
> The reviewer correctly notes that frequency-domain constraints for imperceptible adversarial examples are established in the literature. We acknowledge and agree with this assessment. However, we respectfully argue that our contribution in this dimension is distinct:
>
> 1. Acknowledgment of Prior Work:
> Thank you for the examples of research. We will definitely add explicit citations to Luo et al. (2022) and Qin et al. (2019) in Section 4.4 in the camera-ready version. These works establish the general principle of low frequency constraints. It is important to note that we are not just using someone else's method. Moreover, our frequency masking module is just an additional module that can be added to the main attack without affecting the adversarial perturbation learning process. Thus, our idea of frequency masking is not the main contribution or concept on which the entire proposed method is based.
>
>
>
> 2. Why This Is NIC-Specific Novelty:
> Our paper applies this principle to a new domain (NIC attacks), where its effectiveness is not obvious, because codecs perform lower-level task than other models that have been previously attacked (for example, computer vision models: classifiers and segmentation methods). So, NIC models may unexpectedly destroy high-frequency patterns - codec entropy models target different frequency bands from classifiers. We demonstrate empirically that frequency-domain masking successfully increases the stealth of attack while maintaining its effectiveness.
>
>
> $\textbf{W4:}$
>
> This is a straightforward oversight we will address.
> Perturbation Budget Specification:
> Our work uses $\varepsilon = 7/255$ throughout, which we will make explicit in Section 5 in the camera-ready version. We will also provide the results of additional experiments with other budgets within a couple of days.
>
> We specifically chose a relatively large budget for several reasons:
>
> 1. Interpretability: $\varepsilon = 8/255$ is a commonly used baseline in adversarial robustness research,   making our results comparable to other adversarial attack benchmarks. Our $\varepsilon = 7/255$ falls within this standard range.
>
> 2. Visual Expressiveness: Attacks at this budget level are more interpretable and easier to visualize. Tighter budgets (e.g., 4-5/255) can produce subtle effects that are difficult to see (especially in a paper format), limiting our ability to understand attack mechanisms.
>
> 3. Codec-Appropriateness: Learned codecs are often more robust to tiny perturbations than classifiers, so moderate budgets are necessary to create meaningful attacks. Previous NIC attacks (FTDA, SRDA) use similar or larger budgets.
>
> 4. Fair Baseline Comparison: All compared methods (FTDA, I-FGSM, SRDA) using the same budget to ensure a fair comparison.

---

> ### Author Response · Authors · 2025-11-29
>
> $\textbf{Q1:}$
>
> In our experiments, the number of iterations was selected so that the value of the functions of the attack targets after these iterations already reached a plateau, so there will be no special improvements from more iterations. We also want to note that attacks using only the gradient sign for a step can perform worse with increasing iterations due to low accuracy and discreteness of steps, that is, an increase in the number of iterations is more likely to benefit the normalized direction. We also want to note that in fact, the normalized direction gives a significant increase, this is especially clearly seen on the CED SCORE, for example, in Table 1 with VMAF metrics, thanks to the normalized direction, we managed to raise the CED SCORE from 0.5 to 0.59, which is a significant improvement.
>
> $\textbf{Q2:}$
>
>
> As can be seen in [1], current attack methods have a very low transferability of adversarial samples to other codecs, especially those with different architectures. Since our method was not designed to be highly transferable and does not include the necessary modules (for example, training attacks on multiple codecs simultaneously), it is unlikely that it would have this capability. It is worth noting that increasing transferability often leads to a decrease in the direct effectiveness of the attack, which is why we did not require this from the attack being developed.
>
> $\textbf{Q3:}$
> the answer to the question in the response to $\textbf{W4}$
>
>
> Thank you again for your valuable feedback and time. We appreciate your comments and have taken them into consideration. In our response, we have provided detailed explanations of the non-obvious aspects of our research and fully addressed all the questions you raised. We hope that the information we have provided will help you to better understand the significance and validity of our work. Considering the above, and taking into account that there are no weaknesses in the proposed method, we would like to request that you revise your initial assessment of our research upwards.
>
> [1] Kovalev, Egor, et al. "Exploring adversarial robustness of JPEG AI: methodology, comparison and new methods." arXiv preprint arXiv:2411.11795 (2024).

---

> ### Author Response · Authors · 2025-12-03
>
> As promised, we have included the results of additional experiments with a different attack budget in Appendix A.4.

---

### Official Review · Reviewer_sFxV · 2025-10-31

**Soundness:** 2
**Presentation:** 1
**Contribution:** 1
**Rating:** 2
**Confidence:** 4

**Summary:**

This paper proposes a modular adversarial attack framework tailored for neural image compression (NIC) models. The framework combines global distortion maximization, local artifact amplification, and frequency-domain masking techniques. Experimental results on widely used codecs, such as JPEG AI, demonstrate that the proposed method outperforms existing attacks across multiple compression models (including JPEG AI), producing higher distortion while maintaining imperceptibility.

**Strengths:**

1. The experiments are relatively systematic, involving a comparison of the representative learnable compression methods HiFiC and ELIC with the traditional JPEG method. These experiments confirm that the former provides better defense efficacy, offering insights for researchers aiming to improve adversarial robustness through data preprocessing techniques.
2. The discovery of the "multi-round compression" approach, though simple, provides valuable ideas for lightweight defense strategies.

**Weaknesses:**

1. The motivation behind the use of block-based SSIM difference and pooling to calculate local artifacts in the local optimization module is unclear. Additionally, the parameter PPP has not been subjected to hyperparameter analysis.
2. The design of the evaluation metrics in Section 5.3 is somewhat arbitrary. It lacks integration with subjective assessments or perceptual experiments to guide metric selection.
3. The definitions of the symbols in the formulas are unclear. For example, in Section 4.3, the symbols P, u, and v are not sufficiently explained, making them difficult to understand.
4. The paper lacks image examples of adversarial samples generated using different input methods. There is also an absence of comparative results at different bit rates and across different datasets.

**Questions:**

1. Is attack performance sensitive to block size P and pooling kernel size (e.g., 32)? Has hyperparameter analysis been conducted?
2. In the alternating use of symbolic gradients and normalized gradients, is the frequency of changes fixed or dynamically adjusted? Is there a design to adaptively adjust it based on the loss function?
3. Does your attack framework remain effective when applied to NIC models that have been trained with defense techniques?
4. What is the transferability of adversarial samples generated by your attack framework? For example, if adversarial samples are generated for one NIC model, can they successfully attack another NIC model? Have such experiments been conducted?
5. Why did you choose to use SSIM rather than other perceptual metrics such as LPIPS or DISTS?

---

> ### Author Response · Authors · 2025-11-27
>
> Thank you for your attentive review and constructive comments on the motivation of local optimization, evaluation metrics, and the completeness of experimental validation. Below, we consistently answer all your points and additional questions.
>
> $\textbf{W1:}$
>
>     1. Why block-based SSIM? Following Tsereh et al. (2024), who, after conducting many experiments and relying on research in the field, proposed a highly effective method for detecting neural image compression artifacts based on block-based SSIM difference and pooling. We significantly modified this method for adversarial optimization objective and introduced block decomposition based on pooling to enable controlled multi-artifact generation. Thus, we adapted the method of local artifact detection to our task. This enables:
>
>         a. Detection of multiple artifact regions within a single image
>
>         b. Controlled artifact generation across diverse spatial locations
>
>         c. Better alignment with how humans perceive compression artifacts (spatially clustered)
>
>     2. Hyperparameter P Selection: The parameter P controls the trade-off between artifact localization granularity and computational efficiency. Empirical findings during development showed that P in [50, 100] approximately the same results are obtained. It makes no sense to increase the parameter, since the current size is sufficient for local artifacts to occur, and increasing it will only lead to the loss of some of them. We selected P = 75 as the midpoint providing optimal balance. To completely close this issue, we will provide the results of an additional ablation experiment on the choice of the P parameter within a few days.
>
> $\textbf{W2:}$
>
> We understand the reviewer’s concern about metric selection. However, we argue that our approach is well-justified for this specific application:
>
> Why Subjective Assessment is Impractical Here:
>
>     1. Scale of Evaluation: We test 10 attacks on 12 codecs at 4 bitrates each across 174 images, generating tens of thousands of attack scenarios. A comprehensive subjective study would require hundreds of thousands of human judgments, becoming prohibitively expensive and time-consuming.
>
>     2. Non-Stationarity of Codec Behavior: Different codecs exhibit different vulnerability patterns. JPEG AI’s artifacts differ substantially from ELIC or Ballé 2018 models. Separate subjective assessment for each codec-attack combination would be necessary for fair comparison. Which would also make it more difficult to summarize the results. Therefore, for the general comparison that is required in our case, the final comparison methodology that will lead to honest and understandable results is unclear.
>
>     3. Standardized Metrics in Compression: The compression research community standardly uses PSNR, MS-SSIM, and VMAF for codec evaluation. Using these same metrics for attack evaluation maintains consistency and enables future researchers to build upon our work.
>
> Why Our Metrics Are Principled:
>
>     1. PSNR and MS-SSIM: Established standards in the compression community, directly measure reconstruction quality—the primary attack objective.
>
>     2. VMAF: A modern perceptual quality metric that is able to detect distortions that are not typically detected by previous metrics. Modern neural image compression methods are being honed for this metric: VMAF is fixed in the JPEG AI standard as one of the metrics for evaluating it.
>
>     3. Composite QD-SCORE and CED-SCORE: Rather than arbitrary, these are intentionally designed to:
>
>         a. Average performance across all codecs, avoiding bias toward any single architecture
>
>         b. Combine the assessment of quality reduction and invisibility equally, reflecting the trade-off of effectiveness and attack power.
>
>         c. Provide a single interpretable score for cross-method comparison
>
> $\textbf{W3:}$
>
> Thank you for your feedback. This is a straightforward clarity issue. We will update Section 4.3 with explicit definitions. The $P$ symbol represents the block size in pixels, and $(u, v)$ are the pixel coordinates of the difference map $\Delta$.

---

> ### Author Response · Authors · 2025-11-27
>
> $\textbf{W4:}$
>
> We appreciate this comment and below clarify our experimental design.
>
> #### Visualization Examples:
>
> Our paper does include comprehensive visual results (Fig. 1), showing adversarial images after our attack compared to FTDA and I-FGSM on two representative codecs (JPEG AI and Cheng2020). The reason why we present only the most effective methods is that we are comparing with a large number of attacks on a large number of codecs, so it is much more important for us to fit the most interesting and representative examples. We have added supplementary material with adversarial images before compression (pre-encoding) that correspond to the examples shown in Figure 1.
>
> Suggested Revision: In the camera-ready version, we will include pre-encoding adversarial and original images for Figure 1 and extend it with additional examples.
>
> #### Multi-Bitrate and Multi-Dataset Results:
>
> We explicitly address this:
>
> Multi-Bitrate Evaluation: We test “four different compression rates (or quality levels), covering both low and high bitrate scenarios” (Section 5.1). Rather than reporting results separately for each bitrate, we employ:
> * BSQ-rate Metric (Section 5.4): This is specifically designed to compare compression efficiency across bitrates relative to JPEG2000 baseline
> * CEDM (Compression-Efficiency Drop Metric): Averaged across all bitrates, providing bitrate-invariant performance assessment
>
> Rationale: Different codecs exhibit different compression-quality trade-offs at different bitrates. Reporting results at each bitrate separately would be unwieldy and less interpretable than our composite approach.
>
> Multi-Dataset Coverage: We test on three datasets:
> * KODAK PhotoCD (24 images): Standard benchmark for compression
> * Cityscapes (100 images): Domain-specific urban scenes
> * NIPS 2017 (50 images): Diverse photographic content
>
> Total: 174 images covering diverse domains (photographic, urban, adversarial)
>
> Why Composite Metrics: Reporting separate results for each bitrate-dataset-codec combination would create unwieldy tables (12 codecs ×
> 4 bitrates × 3 datasets = 144 entries per metric) that obscure trends. Our approach prioritizes clarity while maintaining rigor.
>
> Since we are comparing attacks and showing that our method is better than others, we are not so interested in comparing and evaluating the nuances of codecs at a certain bit rate or dataset.
>
> Suggested Revision: We will add a supplementary table showing per-bitrate results for JPEG AI on KODAK as a representative example.
>
> $\textbf{Q1:}$
>
> The kernel size was selected according to the method of Tsereh et al. (2024). For more information, see the response to $\textbf{W1}$.
>
> $\textbf{Q2:}$
>
> The step selection method changes at each iteration, as you can see in Algorithm 2. This approach is also explained in Section 4.2 in the paragraph after the Proposition.
>
> $\textbf{Q3:}$
>
> Many existing methods of defending models from adversarial attacks are not suitable for NICs because they alter the original image, which is not acceptable for the compression task. Additionally, any existing defense approaches are not relevant to JPEG AI until they are incorporated into this standard (while JPEG AI is the most interesting codec to consider and attack). Conducting adversarial training for each codec exceeds the scope of our research, as it is an extremely computationally intensive task necessary only for additional experiment. We would also like to point out that such defense techniques typically only assist against adversarial examples created using undefended models and do not function effectively if the attacker is aware of them during perturbation generation.

---

> ### Author Response · Authors · 2025-11-27
>
> $\textbf{Q4:}$
>
> As can be seen in [1], current attack methods have a very low transferability of adversarial samples to other codecs, especially those with different architectures. Since our method was not designed to be highly transferable and does not include the necessary modules (for example, training attacks on multiple codecs simultaneously), it is unlikely that it would have this capability. It is worth noting that increasing transferability often leads to a decrease in the direct effectiveness of the attack, which is why we did not require this from the attack being developed.
>
> $\textbf{Q5:}$
>
> The reason why we cannot use other perceptual metrics such as LPIPS or DISTS is that these metrics themselves are just as vulnerable to adversarial attacks (as can be seen, for example, in the article [2, 3, 4]). Therefore, if we add them to attack objective, it will lead to the fact that we will hack the metrics themselves and not the codecs. This is also the reason why they are not used as loss functions for training image processing tasks or directly for training NIC models.
>
>
> Thank you again for your valuable feedback and time. We have carefully considered your comments and have provided detailed explanations to support both the proposed method and the assessment methodology, as well as fully disclosed all the issues raised by you. We believe that the information we have provided will help you better understand the significance and quality of our work. Based on this, we ask you to reconsider your initial assessment of our paper and revise it upward.
>
>
> [1] Kovalev, Egor, et al. "Exploring adversarial robustness of JPEG AI: methodology, comparison and new methods." arXiv preprint arXiv:2411.11795 (2024).
>
> [2] Ghildyal, Abhijay, and Feng Liu. "Attacking perceptual similarity metrics." arXiv preprint arXiv:2305.08840 (2023).
>
> [3] Kettunen, Markus, Erik Härkönen, and Jaakko Lehtinen. "E-lpips: robust perceptual image similarity via random transformation ensembles." arXiv preprint arXiv:1906.03973 (2019).
>
> [4] M. Siniukov, D. Kulikov and D. Vatolin, "Applicability limitations of differentiable full-reference image-quality metrics," 2023 Data Compression Conference (DCC), Snowbird, UT, USA, 2023, pp. 1-1, doi: 10.1109/DCC55655.2023.00082.

---

> ### Author Response · Authors · 2025-12-03
>
> We present the results of an additional ablation study on the choice of the P parameter in Appendix A.6.

---

> ### Author Response · Authors · 2025-12-03
>
> As promised, we have included an additional table in Appendix A.7 showing the results of attacks for JPEG AI separately for each bit rate. We would also like to point out that the results for a specific bitrate are shown in Fig. 4 (Section A.2). This figure presents the results for models with the lowest compression ratios in their respective families.

---

### Official Review · Reviewer_mbhN · 2025-11-01

**Soundness:** 2
**Presentation:** 2
**Contribution:** 2
**Rating:** 4
**Confidence:** 3

**Summary:**

This paper proposes an adversarial attack on neural image compression models that operates in two stages: (1) global distortion optimization using MSE, and (2) local artifact amplification using SSIM-based block-wise analysis. The method alternates between signed and normalized gradient directions and includes a frequency-domain masking module for improved stealthiness. Experiments show the attack outperforms prior methods (I-FGSM, FTDA, SRDA) on both traditional NIC models and the modern JPEG AI standard.

**Strengths:**

1. As NIC systems such as JPEG AI are being standardized and deployed, understanding their susceptibility to adversarial perturbations is of significant practical and academic importance. The focus on JPEG AI, a learned codec with real-world relevance—makes this work particularly impactful within the security and image compression communities.

2. The proposed two-stage optimization strategy is conceptually clear and practically effective.

3. The experimental evaluation is thorough and convincing. The authors conduct tests on multiple NIC architectures, including JPEG AI (BOP/HOP variants), and across several datasets such as Kodak, Cityscapes, and NIPS 2017.

**Weaknesses:**

1. While the proposed method is well engineered and empirically effective, its conceptual novelty is limited. The two-stage optimization (MSE → SSIM) and gradient alternation strategies are primarily adaptations or combinations of existing adversarial attack techniques rather than fundamentally new algorithmic ideas. Consequently, the contribution is more engineering-oriented.

2. The paper should include visualizations of the pre-encoding adversarial images (i.e., the perturbed inputs before compression) and their direct comparisons with the original images, since the most straightforward way to verify whether an attack is “imperceptible to the human eye” is to visually inspect the original versus the perturbed (pre-encode) images.

3. The current attack objective maximizes the distortion between the adversarial and original reconstructions (as described in Section Problem Formulation), rather than the distortion with respect to the original image. This seems somewhat strange to me, as it does not directly measure the degradation of visual quality relative to the ground truth.

**Questions:**

Please refer to weaknesses.

---

> ### Author Response · Authors · 2025-11-27
>
> Thank you for your careful and detailed review of our work. We have carefully considered all of your comments and would like to clarify both the technical aspects and the reasoning behind the key decisions made in the article.
>
> $\textbf{W1:}$
>
> We respectfully disagree with this assessment. Our work introduces several truly novel algorithmic and conceptual contributions that significantly differ from existing NIC attack methods:
>
> 1. Normalized Gradient Direction: As far as we know, using standardized gradients (subtracting the mean and dividing by the standard deviation) as an optimization direction for adversarial attacks is the best approach. This differs from previous work in other fields, as the normalized gradient provides a mathematical and principled way to balance pixel-wise sensitivity, addressing the fundamental limitations of sign-based methods that lose information about relative magnitude. Our Proposition 1 offers theoretical justification for this method through Chebyshev inequality bounds on convergence. A method of alternating between a sign-based update and a normalized update has also been proposed and justified.
>
> 2. Two-Stage Optimization for Neural Image Compression: While multi-stage attacks have been proposed in the general adversarial robustness literature, their application to neural image compression is novel. Previous NIC attacks have exclusively used single-objective optimization based on MSE or PSNR. We identify a fundamental characteristic of neural image compression: while reducing global MSE is necessary, visible artifacts are often localized. By explicitly targeting local differences in SSIM in a second stage, we can exploit compression-specific properties that previous attacks overlooked. This approach is not simply a combination of existing methods, but rather an optimization strategy tailored to the specific problem of neural image compression.
>
> 3. SSIM Difference Mapping: The use of difference maps between SSIM(I, C(I)) and SSIM(I*, C(I*)) followed by block-wise maximum pooling to identify and amplify artifact locations is specifically tailored for neural compression. We adapted the artifact detection approach from Tsereh et al. (2024), but significantly modified it for adversarial optimization purposes, and introduced the block decomposition based on pooling to enable controlled multi-artifact generation.
>
> $\textbf{W2:}$
>
> We appreciate this suggestion and understand its value. We have added supplementary material with pre-encode adversarial images that correspond to the examples shown in Figure 1.
>
> $\textbf{Suggested Revision:}$  In the camera-ready version, we will include pre-encoding adversarial and original images for Figure 1 and extend it with additional examples.
>
> $\textbf{W3}$:
>
> It seems to us that the reviewer may have misunderstood the objective of the attack. The reviewer may have been confused by the phrase: "$\hat{I}^*$ is the reconstructed image after compression of $I^\*$", but please note that $I^\*$ is an adversarial image and $\hat{I}^\*$ is its reconstructed version. In fact, the attack objective aims to maximize the distortion between the pre-encoding adversarial image and its reconstructed version, as described in the problem formulation. This can also be seen from the recorded optimization problem corresponding to the attack in the same section:
>
> $\underset{I^* \: |I^* -I|_p \le \varepsilon} {\arg\max} \: d\bigr(I^* ,C(I^*)\bigr)$
>
> Thank you again for your valuable comments and the time spent analyzing the work. Of the three weaknesses, only the first one relates to the scientific content and novelty of the research. We have given a detailed response to this, as well as attached the requested visualizations, and suggested improvements that will be made to the camera-ready version. Considering these explanations and planned improvements, we would like you to reconsider the initial assessment of the paper upwards.

---

### Author Response · Authors · 2025-12-03

Dear ACs,

This summary is provided to assist you in navigating through our rebuttal, highlighting how we have addressed all reviewer comments, clarified our novelty, incorporated necessary changes, and conducted additional experiments.

We present a novel adversarial attack designed specifically for neural image compression (NIC), with additional focus on the JPEG AI standard. Our method combines three key innovations:

1. Normalized Gradient Direction with Alternation: Unlike prior sign-based methods that discard magnitude information, we employ standardized gradients (mean-subtracted, standard-deviation-normalized) alternating with sign-gradients. This principled approach is theoretically justified through Proposition 1 (Chebyshev inequality bounds) and empirically yields significant improvements (for example, CED-SCORE increases from 0.50 to 0.59 on VMAF metrics).

2. Two-Stage NIC-Specific Optimization: While multi-stage attacks exist in general adversarial robustness, we introduce the first optimization strategy tailored to NIC's inherent properties. Stage 1 maximizes global MSE distortion; Stage 2 targets localized artifact amplification using SSIM-difference mapping with block-wise max-pooling. This exploits the fact that compression artifacts are spatially clustered, a property previous NIC attacks overlooked.

3. Frequency-Domain Masking for NIC: We adapt frequency-domain stealth principles to the compression domain, where codec behavior fundamentally differs from classifiers. We demonstrate empirically that this increases imperceptibility without sacrificing attack effectiveness.


### A brief response to the reviewer's concerns.
**On Novelty (mbhN, sFxV, vRSh):** We clarified that our contributions are more than just engineering combinations. The normalized gradient with chaining provides a mathematically sound optimization step supported by bounds. The two-stage formulation directly addresses the specific characteristics of compression, which were absent in previous work. Block-based SSIM mapping with pooling is a novel adaptation for adversarial purposes, that yields to a large number of artifacts, which are of much more varied types compared to previous attacks.



**On Visualization & Clarity (mbhN):** We have added supplementary material (and Appendix A.5) with pre-encoded adversarial images corresponding to the examples in Figure 1. We have also clarified the attack goal: we aim to maximize the distortion between the pre-encoded adversarial input and its reconstruction.


**On Metrics & Evaluation (sFxV):** We justified our metric choices:
* PSNR, MS-SSIM, VMAF are standard in compression research (VMAF is official in JPEG AI).
* Crowdsourced subjective evaluation is impractical at our scale, because it would require hundreds of thousands of human judgments, making it prohibitively expensive and time-consuming. Moreover, there is no established methodology for such subjective comparisons that would produce honest and interpretable results for diverse codec-attack combinations. Though we did conduct our own internal subjective assessments during the method development process, we recognize that this approach is not scalable.
* LPIPS/DISTS cannot serve as loss functions; they are themselves vulnerable to adversarial attacks.
* QD-SCORE and CED-SCORE are principled composites designed to avoid codec bias and balance effectiveness with imperceptibility.


**On Hyperparameters & Completeness (sFxV, vRSh):** We provided:
* Ablation study on P parameter in Appendix A.6 (confirmed P yields consistent results).
* Per-bitrate results for JPEG AI in Appendix A.7.
* Experiments with alternative perturbation budgets in Appendix A.4.
* Explicit attack budget specification in revised Section 5.


**On Theoretical Assumptions (vRSh):** Our Proposition 1 assumes the independence of gradient vector components (not pixels of image), which is realistic for low-level tasks where the gradients resemble uncorrelated high-frequency noise. The Proposition provides worst-case probabilistic bounds ensuring its theoretical soundness - not precise modeling of method.

### Improvements Made:
* Pre-encoding adversarial images added to supplementary material (and Appendix A.5)
* Ablation studies on P and perturbation budget in appendices.
* Per-bitrate breakdown for JPEG AI provided.
* Symbol definitions and attack objective clarified.
* Explicit attack budget specification throughout.

### Conclusion
We have thoroughly and substantively addressed all reviewer concerns. Our contributions - normalized gradient strategies, NIC-tailored two-stage optimization, and compression-aware frequency masking - represent genuine algorithmic and conceptual advances, not mere engineering. All promised revisions and additional experiments are complete and documented in the appendices. We hope that the information we have provided will help you to better understand the significance and validity of our work.

---

### Meta-Review · Area_Chair_Df9z · 2026-01-06

**Summary:**

The three reviewers’ comments are largely consistent and predominantly negative. They raise concerns regarding:
(1) limited novelty of the proposed method, as many of its components already exist in the literature;
(2) insufficient visualization examples illustrating the quality of adversarial images before and after compression, relative to the original benign images;
(3) the lack of justification for the chosen attack objective function;
(4) insufficient justification for the evaluation metrics used in Section 5.3; and
(5) the unclear role, implication, and relevance of the stated proposition.

In addition, the reviewers raise several questions concerning technical details and clarity of presentation.

AC comments. Adversarial attacks on deep neural networks have been extensively studied in the literature. In the context of neural image compression (NIC), the central distinguishing factor lies in the choice of the attack objective function. However, the paper does not provide a principled justification for using the distortion between an adversarial image and its reconstructed version after compression as the attack objective.

Intuitively, a well-founded attack objective should measure the distortion between the original benign image and the reconstructed output of an adversarial image, subject to a small perturbation constraint between the benign and adversarial images. If the distortion between an adversarial image and its reconstructed version does not satisfy distance-like properties, its use as an attack objective becomes questionable.

**Reviewer Concerns:**

Most concerns are still outstanding.

**Reviewer Scores:**

The reviewers would have likely maintained their respective scores.

---

### Decision · Program_Chairs · 2026-01-26

Reject